# Symmetric Wilson Loops beyond leading order

## X. Chen-Lin

Nordita, KTH Royal Institute of Technology and Stockholm University, Roslagstullsbacken 23,
SE-106 91 Stockholm, Sweden
Department of Physics and Astronomy, Uppsala University, SE-751 08 Uppsala, Sweden

* [xinyic@nordita.org](mailto:xinyic@nordita.org)

## Abstract

We study the circular Wilson loop in the symmetric representation of $U(N)$ in $\mathcal{N} = 4$ super-Yang-Mills (SYM). In the large $N$ limit, we computed the exponentially-suppressed corrections for strong coupling, which suggests non-perturbative physics in the dual holographic theory. We also computed the next-to-leading order term in $1/N$, and the result matches with the exact result from the $k$-fundamental representation.

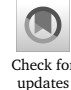

## Contents



# 1  Introduction

The partition function of $\mathcal{N} = 4$ SYM on $S^4$ reduces to a random matrix model with Gaussian unitary ensemble [1–3]. This fact allows exact computation of expectation values of supersymmetric observables, such as circular Wilson loops in various representations of the gauge group $U(N)$ [4].

Motivated by the AdS/CFT correspondence, the regime of interest is when $N$ is large and the 't Hooft coupling, $\lambda = g_{\mathrm{YM}}^2 N$, is strong. For the Wilson loops in $k$-symmetric/anti-symmetric representations, which can be mapped to a system of free bosons/fermions [5], the leading order results matched neatly with their holographic picture, that is, a D3-brane/D5-brane carrying an amount $k$ of the fundamental string charge [6–8]. We recommend [9] for a review.

Concerning the next-to-leading order in strong coupling, [10] computed it recently for the anti-symmetric case, by using the Sommerfeld expansion of the Fermi distribution. However, the derivation for the symmetric case was precluded by the onset of Bose-Einstein condensation. One of our goals here is precisely to compute these subleading corrections in strong coupling for the symmetric case.

The other goal is to compute the next-to-leading order in $1/N$, in order to shed some light on a longstanding discrepancy. That is, the non-planar result in [11] was computed as the correction to the saddle-point that was found by analytic continuation done in [5]. This did not match with the holographic solution derived in [12], which used the spectrum obtained in [13]. In the field theory side, we can say that the path of the saddle was unclear. Therefore, we will avoid analytic continuation, and instead, we will do an honest matrix model computation from scratch, as we did for the $\mathcal{N} = 2^*$ SYM in [14]. Nonetheless, we do expect the non-planar result for the $k$-symmetric representation to match with the $k$-fundamental representation, since Wilson loops in these two representations differ by exponentially-suppressed terms in strong coupling, as shown in [15].

# 2  Wilson loops

A Wilson loop, in the representation $\mathcal{R}$ of the gauge group, $U(N)$ in our case, is defined as the expectation value of the character of the representation, i.e.

$$W = \langle \chi_{\mathcal{R}} \rangle, \quad \chi_{\mathcal{R}} = \mathrm{tr}_{\mathcal{R}} U, \tag{2.1}$$

where $U$ is a path-ordered exponential of the gauge connection $A_\mu$, transported along a close contour $C$:

$$U = \mathrm{P}\exp\left[\oint_C ds \left(i\dot{x}^\mu A_\mu + |\dot{x}|\Phi\right)\right].$$ (2.2)

Here we also have the coupling to one of the scalar fields $\Phi$ of $\mathcal{N}=4$ SYM, in order to make the Wilson loop locally supersymmetric.

For $\mathcal{N}=4$ SYM on $S^4$, the supersymmetric localization technique [3] is applicable to such Wilson loops whose contour lies on the equator of the 4-sphere [1]. In the Euclidean signature, the partition function reduces to a Gaussian matrix model integral:

$$Z = \int d^N a \prod_{i<j}^N (a_i - a_j)^2 \, e^{-\frac{2N}{\lambda}\Sigma_{k=1}^N a_k^2},$$ (2.3)

where the integration variables are the eigenvalues of the vev of the scalar field $\Phi$, i.e.

$$\langle\Phi\rangle = \mathrm{diag}(a_1, \ldots, a_N).$$ (2.4)

## 2.1 Symmetric representation

Our focus will be on Wilson loops in the $k$-symmetric representation, with the Young diagram

$$\mathcal{R} = \overbrace{\boxed{\phantom{xxxx}}}^{k}$$ (2.5)

where $k \sim N$, and we will take $N \to \infty$, so that the ratio $f \equiv k/N$ is kept fixed.

Let us depart from the (Weyl) character formula for the symmetric case, derived in the appendix A, which is

$$\chi_k = \sum_{i=1}^N e^{ka_i} \prod_{j\neq i}^N \frac{1}{1 - e^{a_j - a_i}}.$$ (2.6)

Due to the exponential weight, the main contribution comes from the largest eigenvalue $a_N$ [2]:

$$\chi_k \approx e^{ka_N} \prod_{j=1}^{N-1} \frac{1}{1 - e^{a_j - a_N}}.$$ (2.7)

Furthermore, in the strong coupling limit, the product in (2.7) is exponentially small in $\lambda$, so that the character reduces to the one of the k-wrapped fundamental representation

$$\chi_k \approx e^{ka_N},$$ (2.8)

in agreement with the conclusion of [15].

Let us write the Wilson loop expectation value with (2.7) more explicitly as:

$$W = \frac{1}{Z}\int d^N a \, e^{-N^2 S}, \quad S = S_0 + \frac{1}{N}S_1,$$ (2.9)

---

[1] The compactification on $S^4$ does not really matter in this context because of the conformal invariance of $\mathcal{N}=4$ SYM.

[2] In the appendix F, we comment on the contribution from the rest of the sum in (2.6).

where [3]

$$S_0 = \frac{2}{\lambda}\frac{1}{N}\sum_{i=1}^{N-1}a_i^2 - \frac{1}{N^2}\sum_{i}^{N-1}\sum_{j\neq i}^{N-1}\mathrm{Re}\left[\log(a_j - a_i)\right] \tag{2.10}$$

$$S_1 = \frac{2}{\lambda}a_N^2 - \frac{2}{N}\sum_{i=1}^{N-1}\log(a_N - a_i) - f\,a_N + \frac{1}{N}\sum_{j=1}^{N-1}\log\left(1 - e^{a_j - a_N}\right), \quad f \equiv \frac{k}{N}. \tag{2.11}$$

Since $N$ is large, we will use the saddle-point method to solve (2.9), which yields

$$\log W \approx -N\left(\mathscr{F}_0 + \frac{1}{N}\mathscr{F}_1\right), \tag{2.12}$$

and $\mathscr{F} = \mathscr{F}_0 + \mathscr{F}_1/N$ is the free energy.

## 2.2 Saddle-point equations

Let us use the continuous approximation (for all the eigenvalues except the largest), by introducing the density function:

$$\rho(x) = \frac{1}{N}\sum_{i=1}^{N-1}\delta(x - a_i). \tag{2.13}$$

Note that the normalization condition in this case is:

$$\int_{-c}^{c}dx\,\rho(x) = 1 - \frac{1}{N}, \tag{2.14}$$

implying

$$\int_{-c}^{c}dx\,\rho_0(x) = 1 \quad\text{and}\quad \int_{-c}^{c}dx\,\rho_1(x) = -1, \tag{2.15}$$

if the density is expanded as $\rho = \rho_0 + \rho_1/N$.

Let us also call $A \equiv a_N$. Then, the leading order equations in $N$ are:

$$\fint_{-c}^{c}dy\,\frac{\rho_0(y)}{x - y} = \frac{2x}{\lambda} \tag{2.16}$$

$$\fint_{-c}^{c}dy\,\frac{\rho_0(y)}{A - y} = \frac{2A}{\lambda} - \frac{f}{2} + \frac{1}{2}\int_{-c}^{c}dy\,\frac{\rho_0(y)}{e^{A-y} - 1} \tag{2.17}$$

and the subleading order [4]:

$$\fint_{-c}^{c}dy\,\frac{\rho_1(y)}{x - y} = -\frac{1}{x - A} - \frac{1}{2(e^{A-x} - 1)}. \tag{2.18}$$

In the appendix B, we show a systematic way to solve integral equations and apply it to our equations.

---

[3] For convenience, we put a list of the action and its derivatives in the appendix D. For $S_0$ written here, we used the identity:

$$\sum_{i}^{N-1}\sum_{j=i+1}^{N-1}\log(a_j - a_i) = \frac{1}{2}\sum_{i}^{N-1}\sum_{j\neq i}^{N-1}\mathrm{Re}\left[\log(a_j - a_i)\right].$$

[4] It is not necessarily to expand $A = A_0 + A_1/N$ for our purposes (see footnote 6). It is straightforward to solve it though, for (B.66):

$$\fint_{-c}^{c}dy\,\frac{\rho_1(y)}{A_0 - y} = \frac{2}{\lambda}A_1 \quad\Rightarrow\quad A_1 = -\frac{A_0\,c^2}{2(A_0^2 - c^2)}.$$

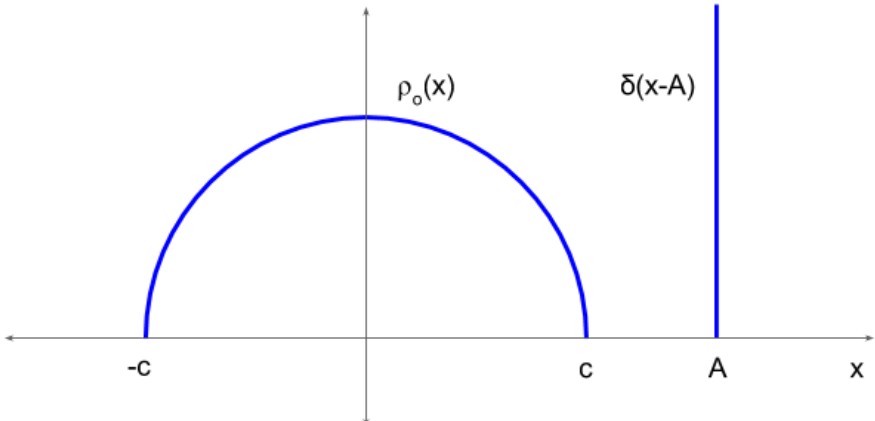

Figure 1: Density distribution in the planar limit.

## 3 Strong coupling correction

In this section, we will work in the planar limit. We will start by reproducing the Drukker-Fiol result [16], and then compute the exponentially-suppressed corrections to it, in strong coupling.

### 3.1 Saddle-point configuration

The solution to the equation (2.16) is the well-known Wigner semi-circle (B.65), that we copy here:

$$\rho_0(x) = \frac{2\sqrt{c^2 - x^2}}{\pi c^2}, \quad c = \sqrt{\lambda}. \tag{3.19}$$

The Bose distribution term in the equation (2.17) is exponentially suppressed in strong coupling, since $A > c$. We thus neglect it for the leading order in strong coupling and, using the semi-circle distribution, it is straightforward to compute $A$, which is

$$A = c\sqrt{\kappa^2 + 1}, \quad \kappa = \frac{cf}{4}. \tag{3.20}$$

We plotted the saddle-point configuration in fig. 1.

#### 3.1.1 Correction to A

The exponentially-small correction to (3.20) (let us denote it by $\delta A$) actually gives higher order corrections to the action, as we shall see later on. We can compute it by expanding (2.17) (with the semi-circle distribution) in $A$ [5] :

$$\int_{-c}^{c} dy\, \frac{\rho_0(y)}{e^{A-y} - 1} = f - \frac{4\sqrt{(A + \delta A)^2 - c^2}}{c^2}. \tag{3.21}$$

Using the geometric sum identity

$$\frac{1}{e^{A-y} - 1} = \sum_{m=1}^{\infty} e^{-(A-y)m}, \tag{3.22}$$

---

[5] Note that we neglected the expansion from the Bose term, since it is exponentially-suppressed.

then, commuting the integration and the sum, and integrating term by term, we end up with the following identity (the series is convergent for $A \geqslant c$):

$$\int_{-c}^{c} dy \frac{\rho_0(y)}{e^{A-y} - 1} = 2 \sum_{m=1}^{\infty} \frac{e^{-Am}}{c\,m} I_1(c\,m), \tag{3.23}$$

where $I_1(z)$ is the modified Bessel function. In the strong coupling limit,

$$2 \sum_{m=1}^{\infty} \frac{e^{-Am}}{c\,m} I_1(c\,m) \approx \frac{2}{\sqrt{2\pi} c^{3/2}} e^{-(A-c)}, \tag{3.24}$$

and we obtain the leading order correction to (3.20):

$$\delta A = -\sqrt{\frac{c\,\kappa^2}{8\pi\,(\kappa^2 + 1)}} e^{-c\left(\sqrt{\kappa^2+1} - 1\right)}. \tag{3.25}$$

## 3.2 The leading order result

The first contribution to the free energy comes from the $1/N$ term in the large $N$ expansion of the action (C.76), since $S_0[\rho_0]$ is canceled by the partition function whose saddle-point distribution is $\rho_0$, hence

$$\mathscr{F}_0 = S_1[\rho_0] - \left[\frac{\delta S_0[\rho_0]}{\delta \rho(x)}\right]_{x=c}, \tag{3.26}$$

where (see the appendix D)

$$S_1[\rho_0] = \frac{2A^2}{c^2} - 2 \int_{-c}^{c} dy\, \rho_0(y) \log(A - y) - f A \tag{3.27}$$

$$\left[\frac{\delta S_0[\rho_0]}{\delta \rho(x)}\right]_{x=c} = 2 - 2 \fint_{-c}^{c} dy\, \rho_0(y) \log(c - y). \tag{3.28}$$

Here, we just dropped the last term in (D.86) (let us call it the Bose term), since it is subleading in strong coupling. We will come back to it in the next subsection.

The integrals can be done using the semi-circle distribution (B.65), and the results are shown in the appendix E. In order to consider how corrections to $A$ (3.20) contribute to the free energy, we expand the free energy result for small $\delta A$, and we arrive to [6]

$$\mathscr{F}_0 = -2\left(\kappa\sqrt{\kappa^2 + 1} + \sinh^{-1}(\kappa) - \frac{2\sqrt{\kappa^2 + 1}}{c^2\kappa} \delta A^2\right), \tag{3.29}$$

which is the well-known result derived by Drukker and Fiol [16] when we drop the subleading $\delta A$ term:

$$\mathscr{F}_0 = -2\left(\kappa\sqrt{\kappa^2 + 1} + \sinh^{-1}(\kappa)\right). \tag{3.30}$$

Our next step is to take into account the ignored Bose term in the action, as in (D.86).

---

[6] The same formula applies to the $1/N$ correction to $A$, and we see clearly that it does not contribute to the next-to-leading order term in large $N$, due to it being quadratic.

### 3.3 Strong-coupling expansion

The strategy to compute the Bose term is to integrate the result (3.23) over $A$, which gives

$$\int_{-c}^{c} dy\, \rho_0(y) \log\left(1 - e^{y-A}\right) = -2 \sum_{m=1}^{\infty} \frac{e^{-Am}}{c\,m^2} I_1(c\,m), \quad A \geqslant c \tag{3.31}$$

where the integration constant is zero, fixed by taking $A$ to infinity. This term, which is an identity, plus the $\delta A$ term in (3.29) constitute the corrections to the Drukker-Fiol solution in the planar limit. In principle, for large $\lambda$, terms of any order can be generated.

Let us derive the first few corrections, by expanding (3.31) in strong coupling. First, the leading contribution comes from the first term of the sum, and then, we expand the modified Bessel function:

$$\int_{-c}^{c} dy\, \rho_0(y) \log\left(1 - e^{y-A}\right) \approx -\frac{2\,e^{-A}I_1(c)}{c} \tag{3.32}$$

$$\approx -\frac{2}{c\sqrt{2\pi c}} e^{-(A-c)} \left(1 - \frac{3}{8c} - \frac{15}{128c^2} + \dots\right). \tag{3.33}$$

The last expansion in the parenthesis can actually be written in a compact way up to $O(c^{-n})$ [7]:

$$\int_{-c}^{c} dy\, \rho_0(y) \log\left(1 - e^{y-A}\right) =$$
$$-\frac{2}{c\sqrt{2\pi c}} e^{-(A-c)} \left(-\frac{\Gamma\left(n-\frac{1}{2}\right)\Gamma\left(n+\frac{3}{2}\right) {}_2F_2\left(1, -n; -n-\frac{1}{2}, \frac{3}{2}-n; -2c\right)}{\pi(2c)^n \Gamma(n+1)}\right). \tag{3.34}$$

Replacing $A$ by (3.20) and $c = \sqrt{\lambda}$, the free energy with the leading exponential corrections is

$$\boxed{\mathscr{F}_0 = -2\left(\kappa\sqrt{\kappa^2+1} + \sinh^{-1}(\kappa) - \sqrt{\frac{2}{\pi}}\frac{e^{-\sqrt{\lambda}\left(\sqrt{\kappa^2+1}-1\right)}}{\lambda^{3/4}}\left[1 - \frac{3}{8\sqrt{\lambda}} - \frac{15}{128\lambda} + O\left(\lambda^{-3/2}\right)\right]\right).}$$
$$\tag{3.35}$$

## 4 Non-planar correction

In this section, we will work in the strong coupling limit. We will compute the next-to-leading order free energy in large $N$, i.e. $\mathscr{F}_1$. In principle, there are three potential sources: the large $N$ expansion of the action, the fluctuations around the saddle-point configuration, and the $SU(N)$ correction. For the latter, we will discuss it separately in the appendix G.

### 4.1 Action

Let us start with the contribution from the action, which is the $1/N^2$ term in (C.76), i.e.

$$\mathscr{F}_{1,S} = \frac{1}{2}\int_{-c}^{c} dx\, \rho_1(x)\frac{\delta S_1[\rho_0]}{\delta\rho(x)} - \frac{1}{2}\left[\int_{-c}^{c} dy\, \rho_1(y)\frac{\delta^2 S_0[\rho_0]}{\delta\rho(x)\delta\rho(y)} + \frac{\delta S_1[\rho_0]}{\delta\rho(x)}\right]_{x=0} \tag{4.36}$$

---

[7] The series expansion for the modified Bessel function used here is taken from: http://functions.wolfram.com/Bessel-TypeFunctions/BesselI/06/02/03/01/.

where the functional derivatives are explicit in the appendix D. The first integral is the same as the integral (E.91), and the second term is

$$\frac{1}{2}\left[\int_{-c}^{c} dy\, \rho_1(y) \frac{\delta^2 S_0[\rho_0]}{\delta\rho(x)\delta\rho(y)} + \frac{\delta S_1[\rho_0]}{\delta\rho(x)}\right]_{x=0} = \log\left(\frac{\sqrt{A+c}-\sqrt{A-c}}{\sqrt{A-c}+\sqrt{A+c}}\right). \tag{4.37}$$

Then, replacing $A$ by (3.20), we get

$$\mathscr{F}_{1,S} = -\log\left(\frac{\left(\sqrt{\kappa^2+1}-\kappa\right)\sqrt{2\kappa\left(\sqrt{\kappa^2+1}+\kappa\right)+1}}{2c\kappa^2}\right). \tag{4.38}$$

## 4.2 Determinant

The saddle-point method gives also the contribution of the quadratic fluctuations around the saddle-point configuration, which is a Gaussian integral that can be integrated:

$$\int_{-\infty}^{\infty} d^N a\, e^{-N^2 S} \approx e^{-N^2 S_*} \sqrt{\frac{(2\pi)^N}{N^{2N}\det\left(S_*''\right)}}. \tag{4.39}$$

Here we formally denoted the Hessian matrix of the action as $S''$, and the subindex $*$ indicates evaluation at the saddle-point configuration. We will skip this latter notation though, for simplicity.

This correction to the partition function must be taken into account as well (see (2.9)), whence the contribution to the free energy is

$$\mathscr{F}_{1,\det} = -\frac{1}{2}\log\left(\frac{\det\left(S_Z''\right)}{\det\left(S''\right)}\right). \tag{4.40}$$

The second derivatives of the action are shown in the appendix D. We will drop the terms with the exponential, since they are subleading in strong-coupling and we are not interested in this now [8]. Notice first that the cross derivatives are of higher order in $1/N$, which allows us to approximate the full determinant using expansion by minors as below:

$$\det(S'') \approx \frac{\partial^2 S}{\partial a_N^2}\det\left(\frac{\partial^2 S}{\partial a_i \partial a_j}\right), \quad i,j \neq N. \tag{4.41}$$

The contribution from the minor is independent of the scaling parameter $\kappa$, and its diagonal elements are actually vanishing, due to the saddle-point equation:

$$\frac{\partial S}{\partial a_k} = 0 \quad \Rightarrow \quad \frac{\partial^2 S}{\partial a_k^2} = 0, \quad k \neq N, \tag{4.42}$$

since the derivative and the finite sum commutes. When $k = N$, we can solve the second derivative by going to the continuous limit (recall $A = a_N$):

$$\frac{\partial^2 S}{\partial A^2} = \frac{4}{c^2 N} + \frac{2}{N}\int_{-c}^{c} \frac{\rho_0(y)}{(A-y)^2}\, dy \tag{4.43}$$

$$= \frac{1}{c^2 N}\frac{4\sqrt{\kappa^2+1}}{\kappa}. \tag{4.44}$$

---

[8] This means we are using the character formula (2.8).

We do not know how to compute the determinant of the minor, neither the determinant from $S_z''$ (which is the same as the minor but one dimension higher), but the dimension analysis suggests their scaling as

$$\det\left(\frac{\partial^2 S}{\partial a_i \partial a_j}\right) \sim \left(\frac{1}{c^2 N}\right)^{N-1}, \quad \det S_Z'' \sim \left(\frac{1}{c^2 N}\right)^{N}.$$

Hence, we conclude that the contribution to the Wilson loop is

$$\mathcal{F}_{1,\,\text{det}} = \frac{1}{2}\log\left(\frac{4\sqrt{\kappa^2 + 1}}{\kappa}\right) + constant, \tag{4.45}$$

where *constant* is not a function of the scaling parameter $\kappa$.

### 4.3 Solution

Summing the contributions (4.38) and (4.45), the subleading order free energy is

$$\boxed{\mathcal{F}_1 = \frac{1}{2}\log\left(\kappa^3\sqrt{\kappa^2 + 1}\right) + \log(4\sqrt{\lambda}) + constant.} \tag{4.46}$$

In order to match this with the exact result from the matrix model computation for the $k$-fundamental representation [17]:

$$\mathcal{F}_{1,\,\square^k} = \frac{1}{2}\log\left(\kappa^3\sqrt{\kappa^2 + 1}\right), \tag{4.47}$$

we see that *constant* must cancel $\log(4\sqrt{\lambda})$. This is consistent, since we neglected the Bose term, meaning we are indeed using the character formula for the $k$-fundamental case (2.8).

## 5 Conclusion

In this paper we used the saddle-point method to compute strong-coupling and non-planar corrections to the leading order solution for the Wilson loop in the symmetric representation.

The strong-coupling corrections come from the Bose statistic term in the character formula (2.7), which is the only term that distinguishes the Wilson loops in the $k$-symmetric representation from the $k$-fundamental representation. We expanded it using geometric series, and for strong coupling, it gives an infinite series of exponentials of negative coupling, where each of these terms has also a power series in $1/\sqrt{\lambda}$. The first leading exponential term is explicitly shown in (3.35), which agrees with the estimate done in [15]. The latter also provided a world-volume interpretation as a disk open string attached to the D-brane. It would certainly be interesting to understand further these non-perturbative corrections in future works.

We would like to comment as well that the fundamental representation also contains non-perturbative corrections [18], when expanding the exact planar result [1] for strong coupling:

$$W_{\text{fund}} = \frac{2}{\sqrt{\lambda}} I_1(\sqrt{\lambda}) \sim e^{\sqrt{\lambda}} + e^{-\sqrt{\lambda}}. \tag{5.48}$$

This is, however, for the Wilson loop, not for the log of the Wilson loop as in our case.

The second goal of the paper was to solve the next-to-leading order term in $1/N$, while remaining at the strong coupling limit. Up to a constant term that we were unable to compute, our solution (4.46) agreed perfectly with the exact $k$-fundamental representation result, which depends only on the scaling parameter $\kappa = k\sqrt{\lambda}/(4N)$. It is consistent from the matrix model

side. However, matching with the holographic dual remains an open question, since the result from the one-loop partition function for the D3-brane computed in [12] is

$$\mathcal{F}_{1,\text{D3}} = \frac{1}{2} \log\left(\frac{\kappa^3}{\sqrt{\kappa^2 + 1}}\right). \tag{5.49}$$

It coincides with the matrix model solution only in the limit $\kappa << 1$, as discussed in [12]. One can also point out corrections from $SU(N)$. Indeed, at the non-planar limit, the traceless condition from $SU(N)$ adds an extra $1/N$ term to the density (see the appendix G). Nonetheless, this does not help solve the mismatch problem. The D3-brane computation should probably be revisited with a better understanding of the D-brane backreaction.

### Acknowledgements

We would like to thank K. Zarembo for useful discussions. This work was supported by the Marie Curie network GATIS of the European Union's FP7 Programme under REA Grant Agreement No 317089, by the ERC advanced grant No 341222 and by the Swedish Research Council (VR) grant 2013-4329.

## A  Weyl character formula for the symmetric representation

Given the generating function $G(\alpha)$, we can derive the character $\chi_k$, i.e.

$$G(\alpha) = \sum_k e^{\alpha k} \chi_k \quad \Rightarrow \quad \chi_k = \int_{C-i\pi}^{C+i\pi} \frac{d\alpha}{2\pi i} e^{\alpha k} G(\alpha). \tag{A.50}$$

For the symmetric representation,

$$G(\alpha) = \prod_{i=1}^{N} \frac{1}{1 - e^{a_i - \alpha}}, \tag{A.51}$$

and $C$ is larger than $a_i$ for all $i$. Assume the eigenvalue set is partially ordered, non-degenerate and finite, i.e. $\{-\infty < a_1 < \ldots < a_N < \infty\}$. Then, we can deform the integration contour to encircle all the eigenvalues, as shown in fig. 2, and use the residue theorem. The contours $C_1$ and $C_2$ cancel each other, and $C_3$ is vanishing. Since the eigenvalues are non-degenerate, they are all single poles, and we obtain (2.6).

## B  Compute the eigenvalue density

Consider the integral equation

$$\fint_a^b dy \, \frac{\rho(x)}{x - y} = F(x). \tag{B.52}$$

Its solution[9] is given by

$$\rho(x) = \fint_a^b \frac{dy}{2\pi} \frac{F(y)}{x - y} \sqrt{\frac{(b - x)(x - a)}{(b - y)(y - a)}} \tag{B.53}$$

---

[9] The reader can check that this is indeed the solution by using the Poincaré-Bertrand transposition formula:

$$\fint_a^b \frac{1}{x - y} \left[ \fint_a^b \frac{f(x,t)}{t - x} dt \right] dx = -\pi^2 f(y,y) + \fint_a^b \left[ \fint_a^b \frac{f(x,t)}{(x - y)(t - x)} dx \right] dt.$$

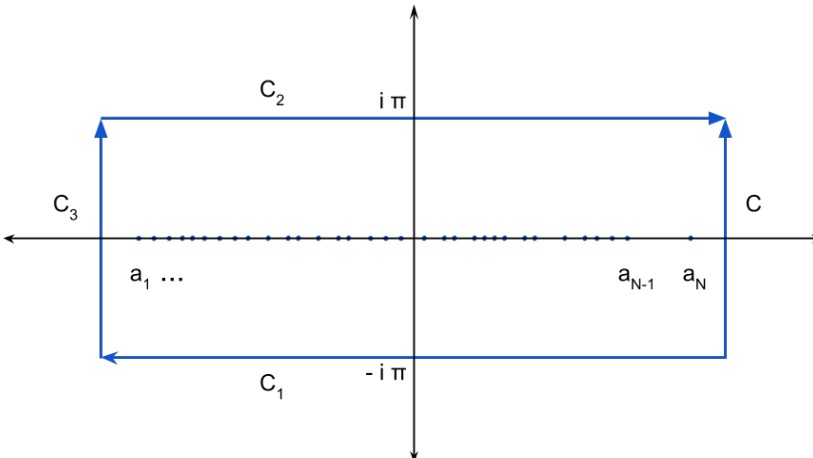

Figure 2: In the $\alpha$ plane, we deform the original integration contour to wrap the eigenvalues that lie on the real axis. The contours $C_1$ and $C_2$ cancel each other, and $C_3$ is vanishing.

with the conditions

$$\int_a^b \frac{dy}{\pi} \frac{F(y)}{\sqrt{(b-y)(y-a)}} = 0 \tag{B.54}$$

$$\int_a^b \frac{dy}{\pi} \frac{yF(y)}{\sqrt{(b-y)(y-a)}} = 1 \tag{B.55}$$

that fix the endpoints $a$ and $b$. The last condition is the normalization condition for the eigenvalue density $\rho(x)$.

We solve the system of (B.53), (B.54) and (B.55) perturbatively in $1/N$. Hence, consider

$$F = F_a + \frac{1}{N}F_b, \quad a = a_0 + \frac{1}{N}a_1, \quad b = b_0 + \frac{1}{N}b_1, \tag{B.56}$$

where, for the case at hand,

$$F_a(x) = \frac{2}{\lambda}x, \quad F_b(x) = -\frac{1}{x-A} + \xi. \tag{B.57}$$

Using $F_a$ only, (B.53) can be solved straightforwardly by the residue theorem, and the result is

$$\rho_a(x) = \frac{2\sqrt{(x-a)(b-x)}}{\pi\lambda}. \tag{B.58}$$

Solving (B.54) and (B.55) at the leading order in $N$ gives

$$c \equiv \sqrt{\lambda} = b_0 = -a_0. \tag{B.59}$$

Expanding $\rho_a$, we get the leading order for the density, and a subleading order term:

$$\rho_a(x) = \frac{2\sqrt{c^2-x^2}}{\pi c^2} + \frac{1}{N}\frac{c(b_1-a_1) + x(a_1+b_1)}{2\sqrt{c^2-x^2}}. \tag{B.60}$$

The next-to-leading order correction to the endpoints are computed using $F_b$ in (B.54) and (B.55), which reduce to

$$a_1 + b_1 = -\frac{c^2}{\sqrt{A^2-c^2}} - c^2\xi \tag{B.61}$$

$$b_1 - a_1 = -\frac{cA}{\sqrt{A^2-c^2}}. \tag{B.62}$$

Then, the $1/N$ correction to $\rho_a$ becomes

$$\rho_{a,1}(x) = -\frac{A+x}{\pi\sqrt{(A^2-c^2)(c^2-x^2)}} - \frac{x\,\xi}{\pi\sqrt{c^2-x^2}} \tag{B.63}$$

We still need to consider the $F_b$ contribution in (B.53), which gives another $1/N$ correction to the density, that is:

$$\rho_{b,1}(x) = \frac{\sqrt{c^2-x^2}}{\pi(A-x)\sqrt{A^2-c^2}}. \tag{B.64}$$

The $1/N$ correction to the density is then the sum of $\rho_{a,1}$ and $\rho_{b,1}$.

## B.1 Solution

In conclusion, we have computed up to the subleading order in $1/N$ for the eigenvalue density. In order to fix the notation, let us denote $\rho = \rho_0 + (\rho_1 + \rho_{1,\xi})/N$, where

$$\rho_0(x) = \frac{2\sqrt{c^2-x^2}}{\pi c^2} \tag{B.65}$$

$$\rho_1(x) = -\frac{\sqrt{A^2-c^2}}{\pi(A-x)\sqrt{c^2-x^2}} \tag{B.66}$$

$$\rho_{1,\xi}(x) = -\frac{x\,\xi}{\pi\sqrt{c^2-x^2}} \tag{B.67}$$

and $c = \sqrt{\lambda}$. When $\xi = 0$, which corresponds to the $U(N)$ case, our solution is in agreement with the result obtained in [17], where the resolvent method was used. If we consider $SU(N)$ instead, the traceless condition fixes the value of $\xi$:

$$\sum_{i=1}^{N} a_i = 0 \quad \Rightarrow \quad N\int_{-c}^{c} dx\, x\,\rho(x) + A = 0 \quad \Rightarrow \quad \xi = \frac{2\sqrt{A^2-c^2}}{c^2}. \tag{B.68}$$

## C  Large N expansion for the action

Given the action $S = S_0 + S_1/N$, let us expand it around $\rho = \rho_0 + \rho_1/N$:

$$\begin{aligned}
S\left[\rho_0 + \frac{1}{N}\rho_1\right] = S_0[\rho_0] &+ \frac{1}{N}S_1[\rho_0] \\
&+ \frac{1}{N}\int_{-c}^{c} dx\,\rho_1(x)\frac{\delta S_0[\rho_0]}{\delta\rho(x)} + \frac{1}{N^2}\int_{-c}^{c} dx\,\rho_1(x)\frac{\delta S_1[\rho_0]}{\delta\rho(x)} \\
&+ \frac{1}{2N^2}\int_{-c}^{c}\int_{-c}^{c} dx\,dy\,\rho_1(x)\rho_1(y)\frac{\delta^2 S_0[\rho_0]}{\delta\rho(x)\delta\rho(y)} + O(N^{-3}).
\end{aligned} \tag{C.69}$$

Using integration by parts, the equation of motion for $S_0$

$$0 = \frac{d}{dx}\frac{\delta S_0[\rho_0]}{\delta\rho(x)}, \tag{C.70}$$

the identity

$$\left[\frac{\delta S_0[\rho_0]}{\delta\rho(x)}\right]_{x=-c} = \left[\frac{\delta S_0[\rho_0]}{\delta\rho(x)}\right]_{x=c}, \tag{C.71}$$

which is a consequence of $\rho_0(x) = \rho_0(-x)$, and the normalization condition for $\rho_1$ in (2.15), we obtain

$$\int_{-c}^{c} dx\, \rho_1(x) \frac{\delta S_0[\rho_0]}{\delta\rho(x)} = -\left[\frac{\delta S_0[\rho_0]}{\delta\rho(x)}\right]_{x=c} \tag{C.72}$$

For the second derivative term, we also use the equation of motion for $S$,

$$0 = \frac{d}{dx}\frac{\delta S[\rho]}{\delta\rho(x)} = \frac{d}{dx}\frac{\delta S_0[\rho]}{\delta\rho(x)} + \frac{1}{N}\frac{d}{dx}\frac{\delta S_1[\rho]}{\delta\rho(x)}, \tag{C.73}$$

then expanding the density $\rho = \rho_0 + \rho_1/N$ and integrating with $\rho_1$, we conclude that

$$\int_{-c}^{c} dy\, \rho_1(y) \frac{\delta^2 S_0[\rho_0]}{\delta\rho(x)\delta\rho(y)} + \frac{\delta S_1[\rho_0]}{\delta\rho(x)} = \text{constant}, \tag{C.74}$$

thus, we can set $x = 0$ to determine the constant. Now, integrating again the above expression with $\rho_1$ yields the following equality:

$$\int_{-c}^{c}\int_{-c}^{c} dx\, dy\, \rho_1(x)\rho_1(y) \frac{\delta^2 S_0[\rho_0]}{\delta\rho(x)\delta\rho(y)} =$$
$$-\int_{-c}^{c} dx\, \rho_1(x) \frac{\delta S_1[\rho_0]}{\delta\rho(x)} - \left[\int_{-c}^{c} dy\, \rho_1(y) \frac{\delta^2 S_0[\rho_0]}{\delta\rho(x)\delta\rho(y)} + \frac{\delta S_1[\rho_0]}{\delta\rho(x)}\right]_{x=0} \tag{C.75}$$

In conclusion, (C.69) reduces to

$$S\left[\rho_0 + \frac{1}{N}\rho_1\right] = S_0[\rho_0] + \frac{1}{N}\left(S_1[\rho_0] - \left[\frac{\delta S_0[\rho_0]}{\delta\rho(x)}\right]_{x=c}\right)$$
$$+ \frac{1}{2N^2}\left(\int_{-c}^{c} dx\, \rho_1(x)\frac{\delta S_1[\rho_0]}{\delta\rho(x)} - \left[\int_{-c}^{c} dy\, \rho_1(y)\frac{\delta^2 S_0[\rho_0]}{\delta\rho(x)\delta\rho(y)} + \frac{\delta S_1[\rho_0]}{\delta\rho(x)}\right]_{x=0}\right)$$
$$+ O(N^{-3}). \tag{C.76}$$

# D  Action and derivatives

For convenience, we write a list of the action, $S = S_0 + S_1/N$, and its derivatives, for the Wilson loop in the $k$-symmetric representation of $U(N)$. Both the discrete version and the continuous approximation will be relevant for many computations in the paper.

### D.1 Discrete

$$S_0 = \frac{2}{\lambda N} \sum_{n=1}^{N-1} a_n^2 - \frac{2}{N^2} \sum_{i=1}^{N-1} \sum_{j=i+1}^{N-1} \log\left(a_j - a_i\right) \tag{D.77}$$

$$S_1 = \frac{2}{\lambda} a_N^2 - \frac{2}{N} \sum_{i=1}^{N-1} \log\left(a_N - a_i\right) - f a_N + \frac{1}{N} \sum_{j=1}^{N-1} \log\left(1 - e^{a_j - a_N}\right) \tag{D.78}$$

$$\frac{\partial S}{\partial a_k} = \frac{4}{\lambda N} a_k - \frac{2}{N^2} \sum_{j \neq k}^{N} \frac{1}{a_k - a_j} + \frac{1}{N^2} \frac{1}{e^{a_N - a_k} - 1}, \quad k \neq N \tag{D.79}$$

$$\frac{\partial S}{\partial a_N} = \frac{4}{\lambda N} a_N - \frac{2}{N^2} \sum_{j=1}^{N-1} \frac{1}{a_N - a_j} - \frac{f}{N} - \frac{1}{N^2} \sum_{j=1}^{N-1} \frac{1}{e^{a_N - a_j} - 1} \tag{D.80}$$

$$\frac{\partial^2 S}{\partial a_k^2} = \frac{4}{\lambda N} + \frac{2}{N^2} \sum_{j \neq k}^{N} \frac{1}{\left(a_k - a_j\right)^2} + \frac{1}{N^2} \frac{e^{a_N - a_k}}{\left(e^{a_N - a_k} - 1\right)^2}, \quad k \neq N \tag{D.81}$$

$$\frac{\partial^2 S}{\partial a_N^2} = \frac{4}{\lambda N} + \frac{2}{N^2} \sum_{j=1}^{N-1} \frac{1}{\left(a_k - a_j\right)^2} + \frac{1}{N^2} \sum_{j=1}^{N-1} \frac{e^{a_N - a_j}}{\left(e^{a_N - a_j} - 1\right)^2} \tag{D.82}$$

$$\frac{\partial^2 S}{\partial a_l \partial a_k} = -\frac{2}{N^2 \left(a_k - a_l\right)^2}, \quad l \neq k \neq N \tag{D.83}$$

$$\frac{\partial^2 S}{\partial a_N \partial a_k} = -\frac{2}{N^2 \left(a_k - a_N\right)^2} - \frac{1}{N^2} \frac{e^{a_N - a_k}}{\left(e^{a_N - a_k} - 1\right)^2} \tag{D.84}$$

### D.2 Continuous

$$S_0[\rho] = \frac{2}{\lambda} \int_{-c}^{c} dy\, \rho(y) y^2 - \fint_{-c}^{c} dy\, \rho(y) \int_{-c}^{c} dz\, \rho(z) \mathrm{Re}\left[\log(z - y)\right] \tag{D.85}$$

$$S_1[\rho] = \frac{2}{\lambda} A^2 - f A - 2 \int_{-c}^{c} dy\, \rho(y) \log(A - y) + \int_{-c}^{c} dy\, \rho(y) \log(1 - e^{y - A}) \tag{D.86}$$

$$\frac{S_0[\rho]}{\delta \rho(x)} = \frac{2}{\lambda} x^2 - \fint_{-c}^{c} dy\, \rho(y) \log((y - x)^2) \tag{D.87}$$

$$\frac{\delta S_1[\rho]}{\delta \rho(x)} = -2 \log(A - x) + \log(1 - e^{x - A}) \tag{D.88}$$

$$\frac{\delta^2 S_0[\rho]}{\delta \rho(x) \delta \rho(y)} = \begin{cases} -\log\left((y - x)^2\right) & \text{if } x \neq y \\ 0 & \text{if } x = y \end{cases} \tag{D.89}$$

## E    Integrals

The integrals below are used to compute the on-shell action. They are valid for $A > c > 0$ and $c > x > -c$, except for the one with $\rho_0$, which is also valid when $A = c$. The explicit expressions for the densities can be found in the appendix B.1.

$$\int_{-c}^{c} dy\, \rho_0(y) \log(A - y) = -\frac{1}{2} - \frac{A\left(\sqrt{A^2 - c^2} - A\right)}{c^2} + \log\left(\frac{\sqrt{A^2 - c^2} + A}{2}\right) \tag{E.90}$$

$$\int_{-c}^{c} dy\, \rho_1(y) \log(A - y) = \frac{1}{2} \log\left(\frac{2A\left(\sqrt{A^2 - c^2} + A\right) - c^2}{4\left(A^2 - c^2\right)^2}\right) \tag{E.91}$$

$$\fint_{-c}^{c} dy\, \rho_1(y) \mathrm{Re}\left[\log(y - x)\right] = -\log(A - x) - \log\left(\frac{\sqrt{A + c} - \sqrt{A - c}}{\sqrt{A - c} + \sqrt{A + c}}\right) \tag{E.92}$$

$$\int_{-c}^{c} dy\, \rho_{1,\xi}(y) \log(A - y) = \left(A - \sqrt{A^2 - c^2}\right) \xi \tag{E.93}$$

# F  Exponentially-small corrections in N

Let us write the character (2.6) as

$$\chi_k = \chi_{k,1} + \chi_{k,\mathrm{rest}}, \tag{F.94}$$

where

$$\chi_{k,1} = e^{ka_N} \prod_{j=1}^{N-1} \frac{1}{1 - e^{a_j - a_N}} \tag{F.95}$$

$$\approx e^{N\left(fA - \int_{-c}^{c} dy\, \rho(y) \log\left(1 - e^{y - A}\right)\right)} \tag{F.96}$$

and

$$\chi_{k,\mathrm{rest}} = \sum_{i=1}^{N-1} e^{ka_i} \prod_{j \neq i}^{N} \frac{1}{1 - e^{a_j - a_i}} \tag{F.97}$$

$$\approx N \int_{-c}^{c} \rho(x)\, e^{N\left(fx - \fint_{-c}^{c} \rho(x) \log\left(1 - e^{y - x}\right) - \log\left(1 - e^{A - x}\right)\right)}. \tag{F.98}$$

We used the equilibrium distribution (B.65) and (3.20) for the approximations above.

Thus, the Wilson loop, i.e. the expectation value of the character, can be written as

$$W = W_1 + W_{\mathrm{rest}} \tag{F.99}$$

$$= W_1 \left(1 + N \int_{-c}^{c} \rho(x) e^{-N\Gamma(x)}\, dx\right), \tag{F.100}$$

where

$$\Gamma(x) \equiv f(A - x) + \fint_{-c}^{c} \rho(y) \log\left(\frac{1 - e^{y - x}}{1 - e^{y - A}}\right) dy + \log\left(1 - e^{A - x}\right). \tag{F.101}$$

If $\Gamma(x) > 0$, then at large $N$, the integral must be much smaller than 1:

$$\log W - \log W_1 = \log\left(1 + N \int_{-c}^{c} \rho(x) e^{-N\Gamma(x)}\, dx\right) \tag{F.102}$$

$$\approx N \int_{-c}^{c} \rho(x) e^{-N\Gamma(x)}\, dx \tag{F.103}$$

$$\approx e^{-N\Gamma_{\mathrm{eff}}(x_*)}, \tag{F.104}$$

where in the last step, the saddle-point approximation is used.

We check numerically that $\Gamma(x) > 0$ for all $x$ in the interval $[-c, c]$ when $c$ is large enough, see the plots in fig. 3. Since we are in the strong-coupling regime, the correction from $\chi_{k,\mathrm{rest}}$ is certainly exponentially suppressed for large $N$.

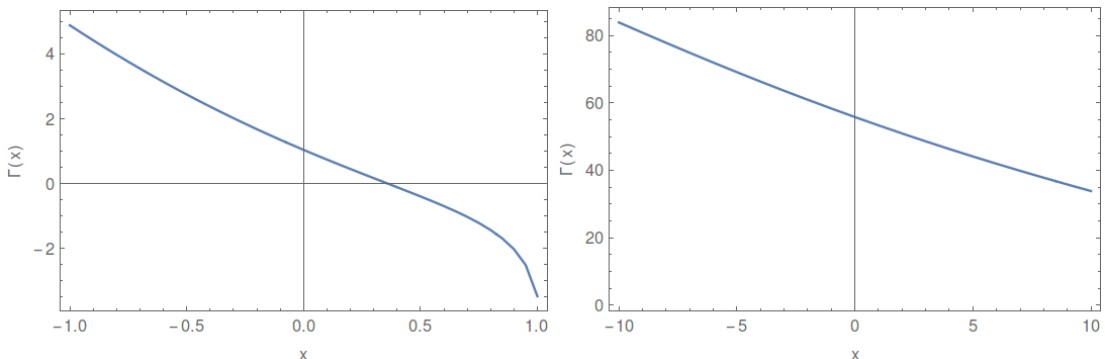

Figure 3: The left plot is when $c = 1$, and $\Gamma(x)$ can have negative values. The right plot is when $c = 10$, and $\Gamma(x)$ is positive for the eigenvalue interval. In both plots, $f = 1$ is used.

## G SU(N) correction

The body of this paper has focused on the $U(N)$ gauge group. Nevertheless, usual holography is referred to $SU(N)$, though there are arguments [19] favoring $U(N)$. Since our computation is quite straightforward, let us compute explicitly the correction of $SU(N)$ to the subleading order in large $N$. All we need to do it is to add a Lagrange multiplier term to the action

$$S_\xi = \frac{2\xi}{N^2} \sum_{i=1}^{N} a_i, \tag{G.105}$$

in order to enforce the traceless constraint:

$$\frac{dS_\xi}{d\xi} = 0 \quad \Rightarrow \quad \sum_{i=1}^{N} a_i = 0. \tag{G.106}$$

The saddle-point equations also get modified, by adding $\frac{\partial S_\xi}{\partial a_i} = \frac{2\xi}{N^2}$ to the derivatives in D. Hence, the $SU(N)$ correction to the Wilson loop enters only through the density, see B.1. The contribution to the log of the Wilson loop is exactly the integral (E.93), where $A$ should be replaced by (3.20) and $\xi$ by (B.68), and we end up with the free energy

$$\mathscr{F}_{1,\xi} = -2\kappa\left(\sqrt{1+\kappa^2} - \kappa\right). \tag{G.107}$$

We see that this addition to our $U(N)$ result (4.46) does not match either with the holographic solution (5.49).

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
