# Peer review of "Symmetric Wilson Loops beyond leading order"

_SciPost Physics, doi:SciPost Phys. 1, 013 (2016)_

## Round 2 · Referee Report · Anonymous · 2016-11-25

Strengths

1. Novel results that might help solving a longstanding discrepancy.

2. Technically strong, but clearly presented.

Weaknesses

1 As explained in the requested changes, the main results of the paper ought to be compared in more detail with existing literature.

Report

This paper deals with the computation of the vacuum expectation value of 1/2 BPS circular Wilson loop in the k-th symmetric representation of ${\cal N}=4$ SU(N) SYM. This quantity can in principle be computed exactly, using the technique of supersymmetric localization.

One of the main applications of carrying out such computation is to perform precision tests of the AdS/CFT duality, since it is known that on the string side, these operators correspond to certain D3 branes. There has been vigorous efforts to extend the original match of the leading terms, to compute and compare $e^{-\sqrt{\lambda}}$ and $1/N$ corrections on both sides of the duality. So far, however, there are unsolved discrepancies.

The paper under consideration deals only with field theory computations, that boil down to matrix model computations. The paper provides two main results for the vev of the Wilson loop in the symmetric representation. First, in the planar limit and at strong coupling, $e^{-\sqrt{\lambda}}$ corrections. Second, it presents the leading 1/N correction at strong coupling.

Requested changes

1.- The author ought to comment on the similarity of the main result in section 3, namely eq. (3.35), with the results of reference [14]. Furthermore, in the Conclusion it is mentioned that the subleading term in (3.35) hints at world-volume instantons. As a matter of fact, reference [14] attempted to provide a world-volume derivation of such terms. This should be mentioned and commented in the paper.

2.- As for the main result in section 4, eq. (4.46), it is worth stating explicitly in the paper that this result disagrees, not only with the string computation in [11], but also with a previous matrix model computation, presented in reference [13]. This information is buried, quite inderectly, in footnote (1), but deserves to be mentioned explictly and expanded upon in the main text.

  • validity: high
  • significance: good
  • originality: good
  • clarity: good
  • formatting: good
  • grammar: good

Author:  Xinyi Chen-Lin  on 2016-12-02  [id 78]

(in reply to Report 1 on 2016-11-25)
Category:
reply to objection

Here I thank the referee for the comments. 1. Indeed, I shall comment on what is done in [14] for the world-volume instantons. 2. [13] computed the saddle-point correction by extending the analysis done in [9], which I commented that the analytical continuation there is unclear. To compute the correction, one needs to know well the path where the saddle-point lies, so [13] is most likely wrong.

---

## Round 2 · Referee Report · Anonymous · 2016-11-29

Strengths

1. The results are new.
2. The method is quite reliable.
3. The results are useful.

Weaknesses

1. The results are not quite suggestive.

Report

In this paper, the author considers the expectation value of the 1/2 BPS circular Wilson loop in the symmetric representation. This expectation value is known to reduce to the calculation of the Gaussian matrix model. The author analyzes this Gaussian matrix model. The leading term in the large N and strong coupling scaling limit is first calculated by Drukker and Fiol. It is known to agree with the D3-brane calculation. In this paper, the author calculated some sub-leading correction terms both in strong coupling and large N scaling limit. The method used here is quite reliable and the results seem to be correct. These results will be compared to the AdS side in the future, and the AdS/CFT correspondence in the sub-leading correction will be discussed. For these reasons, I would recommend publishing this paper after a few minor corrections.

Requested changes

1. In left-hand side of eq. (3.21), $\delta A$ is omitted without any explanation. I found it does not change the form (3.25) later, but I was a little bit confused when I first saw eq. (3.21). Thus to make the explanation more clear, some comment on omitting $\delta A$ should be added around (3.21).

2. The first sentence of Subsection 4.3 is
Summing the contributions (4.40) and (4.45), ...
I guess this should be
Summing the contributions (4.38) and (4.45), ...

  • validity: high
  • significance: good
  • originality: good
  • clarity: high
  • formatting: good
  • grammar: good

Author:  Xinyi Chen-Lin  on 2016-12-02  [id 77]

(in reply to Report 2 on 2016-11-29)
Category:
reply to objection

First, let me thank the referee for the report.
As for the requested changes, the point 2 is indeed a mistake, so I will correct it. For point 1, (3.21) only keeps the leading contribution in $\delta A$, since the LHS of the equation gives an exponentially suppressed contribution in $\delta A$. Maybe I can write it a bit more clearly.

---

## Editorial Decision

published